# Age-dependent diastolic heart failure in an in vivo *Drosophila* model

**Matthew P Klassen[1]\*, Christian J Peters[1], Shiwei Zhou[1], Hannah H Williams[1], Lily Yeh Jan[1,2], Yuh Nung Jan[1]\***

[1]Department of Physiology, Howard Hughes Medical Institute, University of California, San Francisco, San Francisco, United States; [2]Department of Biochemistry and Biophysics, University of California, San Francisco, San Francisco, United States

**Abstract** While the signals and complexes that coordinate the heartbeat are well established, how the heart maintains its electromechanical rhythm over a lifetime remains an open question with significant implications to human health. Reasoning that this homeostatic challenge confronts all pulsatile organs, we developed a high resolution imaging and analysis toolset for measuring cardiac function in intact, unanesthetized *Drosophila melanogaster*. We demonstrate that, as in humans, normal aging primarily manifests as defects in relaxation (diastole) while preserving contractile performance. Using this approach, we discovered that a pair of two-pore potassium channel (K2P) subunits, largely dispensable early in life, are necessary for terminating contraction (systole) in aged animals, where their loss culminates in fibrillatory cardiac arrest. As the pumping function of its heart is acutely dispensable for survival, *Drosophila* represents a uniquely accessible model for understanding the signaling networks maintaining cardiac performance during normal aging.

**\*For correspondence:** matthew.
klassen@ucsf.edu (MPK);
YuhNung.Jan@ucsf.edu (YNJ)

**Competing interests:** The
authors declare that no
competing interests exist.

**Reviewing editor:** Talila Volk,
Weizmann Institute of Science,
Israel

## Introduction

During an average human life span, the heart will undergo 2.8 billion cycles of contraction and relaxation, wherein its four chambers rhythmically beat in a precisely choreographed sequence to efficiently circulate the blood. However, despite our understanding of the molecular basis of the heartbeat (*Bers, 2008*; *Monfredi et al., 2013*; *Solaro, 2010*), societies must confront an increasing failure of this process, which represents an intersection between normal aging and deleterious environmental and genetic factors (*Yancy et al., 2013*). Previous research has demonstrated that many biophysical properties of the heart are altered over the course of a lifetime (*Carrick-Ranson et al., 2012*; *Cheng et al., 2009*; *Lakatta et al., 2014*). These changes exist structurally, with myocardial fibrosis and arterial stiffening playing a leading role, but also manifest functionally, with alterations in the electrical waveform and calcium handling. However, whether these changes reflect normal senescence or compensatory attempts to maintain function remains an intense area of investigation.

The maintenance of cellular and organismal physiology, termed homeostasis, is essential for all living organisms. To uncover the mechanisms that maintain cardiac function, it has been necessary to develop a systems perspective (*Kohl et al., 2010*). These efforts have revealed that the heart's rhythmic behavior is not uniform but exhibits variations that precisely match local mechanical needs (*Brutsaert, 1987*; *Schram et al., 2002*). Alterations in the heart's rhythm also occur during normal aging, where the action potential is prolonged in an apparent effort to maintain the intracellular calcium dynamics necessary for contraction (*Janczewski et al., 2002*). Such electrical remodeling is also observed in chronic heart failure, atrial fibrillation and in several genetic and pharmacological models, suggesting that ion channels are under homeostatic regulatory control (*Nattel et al., 2007*; *Rosati and McKinnon, 2004*; *Schmitt et al., 2014*). These observations raise the possibility that the

heart monitors its own efficacy and adaptively remodels its electromechanical tone. However, significant implications to the pathophysiology of heart disease notwithstanding, the extended lifetimes and complexity of traditional mammalian models, as well as the necessity of their cardiovascular system for organismal survival, have confounded our ability to examine these potential homeostatic responses.

In seeking to develop a more tractable model for understanding this process, we reasoned that all electromechanical biological oscillators must defend themselves against intrinsic and environmental instability and compensate for natural senescence if they are to maintain their efficacy over time. The first primordial cardiomyocytes likely originated in Bilateria, and the transcription factors *tinman* (*Nkx2.5* in mammals), *hand* and *mef2* represent a conserved cardiogenic program linking the evolution of Protostome and Deuterostome hearts (*Bishopric, 2005*). Despite having an open circulatory system that reverses the direction of flow periodically (*Wasserthal, 2007*) and a greatly simplified architecture (*Rotstein and Paululat, 2016*), *Drosophila* has proven to be of significant utility in the study of cardiac development and physiology, revealing mechanisms of cardiogenesis and heart function that are highly conserved with mammals (*den Hoed et al., 2013*; *Frasch, 2016*; *Monnier et al., 2012*; *Neely et al., 2010*). Consequently, *Drosophila* serves as a uniquely high-throughput 'pioneer' genetic model for uncovering conserved pathways involved in cardiomyocyte development and function.

A variety of imaging modalities have investigated the molecular mechanisms underlying cardiac function in *Drosophila* (*Ocorr et al., 2014*). Most notably, an in situ preparation has been used to isolate the intrinsic regulators of cardiac performance in a defined physiological solution and in the absence of neuronal input. A complementary approach would be to monitor heart function in intact animals, where the full suite of intrinsic, environmental and homeostatic processes regulating the heart would be accessible to investigation, with minimal decay in cardiac performance for hours after preparation. While simple transmitted or reflected luminance measures can robustly monitor the rhythm of the heart in vivo, one cannot accurately measure heart wall displacement using this approach. Optical coherence tomography has been utilized to measure heart rhythm and displacement in vivo, and has successfully uncovered several novel genes affecting cardiac function (*Choma et al., 2006*; *Wolf et al., 2006*; *Li et al., 2013*; *Alex et al., 2015*). However, increasing heart wall contrast relative to the lumen and surrounding tissues could yield further improvements in our ability to assess cardiac function in the intact animal.

In this study, we present a high resolution fluorescence imaging and analysis toolset for measuring cardiac function in intact, unanesthetized *Drosophila melanogaster*. Using this platform, we demonstrate that, as in humans, normal aging manifests primarily as defects in relaxation (diastole) while preserving contractile performance, suggesting a conserved susceptibility to aging-related declines in the electrical, biochemical or structural processes facilitating relaxation. We also uncover a critical role for a heteromeric two-pore potassium channel in maintaining cardiac rhythmicity during aging, which appears dispensable for heart function early in life but is critical for preventing fibrillatory cardiac arrest in aged animals. We propose that the robustness, speed and resolution of this in vivo platform will significantly increase the utility of *Drosophila* in understanding conserved mechanisms of cardiac aging and homeostasis.

## Results

### Imaging cardiac performance in intact, unanesthetized *Drosophila*

We developed a fluorescence-based approach for imaging the heart in intact, unanesthetized *Drosophila*. Briefly, the red fluorophore tdTomato was transgenically expressed in working cardiomyocytes using a newly discovered heart enhancer R94C02 (*Tables 1* and *2*). Intact flies were attached to coverslips using ultraviolet-activated optical cement and placed in the optical path (*Figure 1A*). Excitation light was spatially limited to the region of the dorsal abdomen containing the second and third chambers of the heart using a digital-micromirror projector, with the fluorescence emitting back through the cuticle captured on a sCMOS camera operating at 120 frames per second (*Figure 1B*).

Using this in vivo preparation and an automated feature detection algorithm based on maximal contrast, we unambiguously tracked the heart wall across the cardiac cycle in intact animals

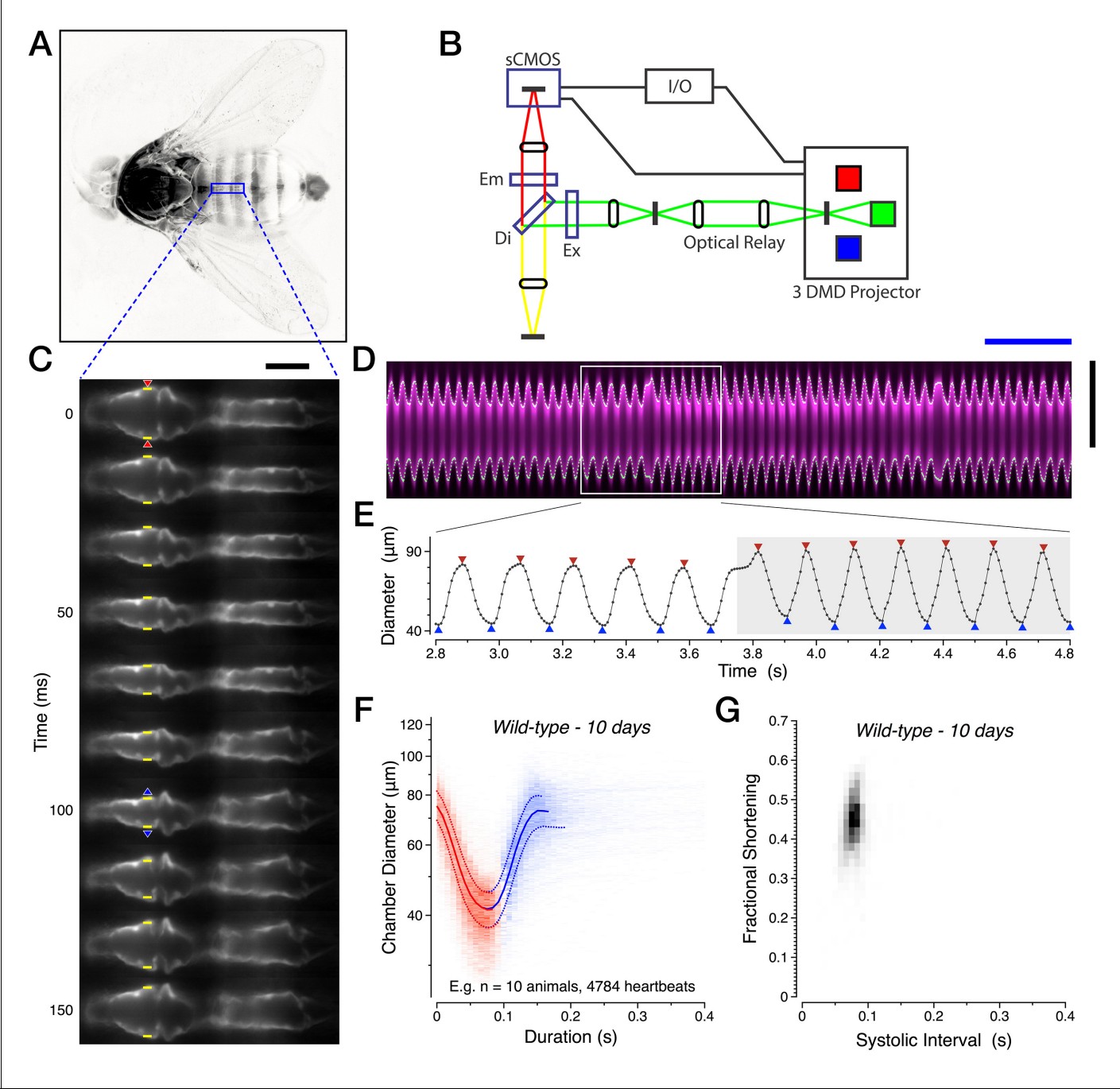

**Figure 1.** Imaging cardiac performance in intact, unanesthetized *Drosophila*. (**A**) Full view of the intact preparation, with imaging region of interest (blue). The head and legs are freely moving, while the wings, dorsal thorax and dorsal abdomen are affixed to the coverglass with optical cement. (**B**) Electro-optical diagram of the imaging system. (**C**) Single anterograde heartbeat at half frame-rate with heart wall position (yellow), initiation of contraction (red triangles) and of relaxation (blue triangles) calls. (**D**) Associated YT kymograph (magenta) with heart wall detection (white dots). (**E**) Corresponding digitization, segmented into anterograde (white) and retrograde (grey) heartbeat epochs. The triangles denote the initiation (red) and end (blue) of contractions. (**F**) Two-dimensional probability map of heart chamber diameter and heartbeat duration with median +/− quartile overlay for systole (red) and diastole (blue). (**G**) Two-dimensional probability map of fractional shortening and systolic interval. All data in this figure are representative and from the 10-day-old $w^{1118}$ wild-type dataset. See Materials and methods for all functional parameter definitions and their derivation. Scale bars: (black vertical) 75 µm, (blue horizontal) 1 s. See also *Video 1*.

The following figure supplement is available for figure 1:

**Figure supplement 1 .** Further characterization of the intravital imaging methodology.

**Video 1.** Heartbeat visualization, digitization and segmentation. One-third speed video of the 10 day adult female displayed in *Figure 1C–E*, with heart wall position calls (yellow) and the attending transformation into heart chamber diameter as a function of time in a 1 s streaming window. The initiation and end of each contraction are specified by a red and blue triangle, respectively. Note the periodic reversal in the direction of heartbeat peristalsis.

(*Figure 1C,D*). We then developed a segmentation algorithm that converted this digital representation of chamber diameter over time into discrete contraction and relaxation events (*Figure 1C–E*, *Video 1*). This segmentation allowed us to derive a diverse set of heart functional parameters, including estimates of cardiac output and stroke volume. We refer the reader to the Materials and methods section for a detailed explanation of the algorithms and formulas used.

In wild-type animals, heartbeats are very consistent and can be visualized *en masse* by assembling a two-dimensional probability map of chamber diameter across the cardiac cycle (*Figure 1F*) or fractional shortening versus the systolic interval (*Figure 1G*), which measures the percentage reduction in chamber diameter during a contraction. Both visualizations demonstrated that the mechanical rhythm of the heart is reproducible between animals. This consistency is also evident in the four primary measurements of the cardiac cycle: the systolic interval, diastolic interval, end systolic diameter, and end diastolic diameter. The standard deviations for these four measurements are between 7% and 11% for our wild-type dataset of young animals between 10 and 30 days of age (n.s., Kruskal-Wallis one-way ANOVA followed by Dunn's multiple comparisons test, n = 68), demonstrating that this in vivo preparation is highly reproducible.

We next performed a series of control experiments to assess the stability of the preparation. The excitation light intensity utilized did not appreciably alter the heartbeat waveform but at higher light intensities, the kinetics of contraction and relaxation accelerated significantly (*Figure 1—figure supplement 1A,B*, $R^2 = 0.93$). Furthermore, the preparation was stable for at least two hours, with cardiac output maintained even after 19 hr in flies kept hydrated overnight (*Figure 1—figure supplement 1C,D*), establishing that our approach is not subject to meaningful variability associated with decay in the health of the preparation.

## Normal aging manifests as diastolic dysfunction while preserving contractile performance

While modifiable risk factors including elevated blood pressure, tobacco use, abnormal blood sugar levels, physical inactivity and obesity strongly exacerbate the incidence of heart disease (*Yancy et al., 2013*), aging-related changes in the structure and physiology of the heart also influence disease progression (*Lakatta et al., 2014*). Increases in arterial load, ventricular hypertrophy and diastolic dysfunction are well established in healthy aging hearts (*Carrick-Ranson et al., 2012*; *Cheng et al., 2009*; *Lakatta et al., 2014*). At the molecular level, animal models have revealed clear alterations in action potential duration and calcium handling with age (*Feridooni et al., 2015*). However, the relative contribution of physiological remodeling in cardiomyocytes to declines in cardiac performance has proven difficult to deconvolve from defects arising from arterial overload and structural hypertrophy.

To establish whether our *Drosophila* platform might represent a simplified model to study cardiomyocyte senescence in vivo, we aged female flies to determine how heart function would change over time. Previous research using an in situ dissected preparation in *Drosophila* has demonstrated progressive declines in cardiac rhythmicity during the first five weeks of life (*Ocorr et al., 2007*). In our in vivo preparation, the heart rate and cardiac performance of flies was remarkably stable for up to 30 days of age in two 'wild-type' genetic backgrounds, $w^{1118}$ and *Canton S*, which we subsequently grouped for further analysis (6.9% maximal variance in heart rate between these two groups at 10, 20 and 30 days of age, n.s., Kruskal-Wallis one-way ANOVA followed by Dunn's multiple comparisons test). At 50 days of age, flies displayed a small decrease in heart rate, (*Figure 2A*, $R^2 = 0.84$) which primarily reflected a prolongation of the systolic interval (*Figure 2—figure supplement 1A*). Only after 50 days of age did the diastolic interval lengthen (*Figure 2—figure supplement 1B*) and cardiac output per second decline (*Figure 2B*).

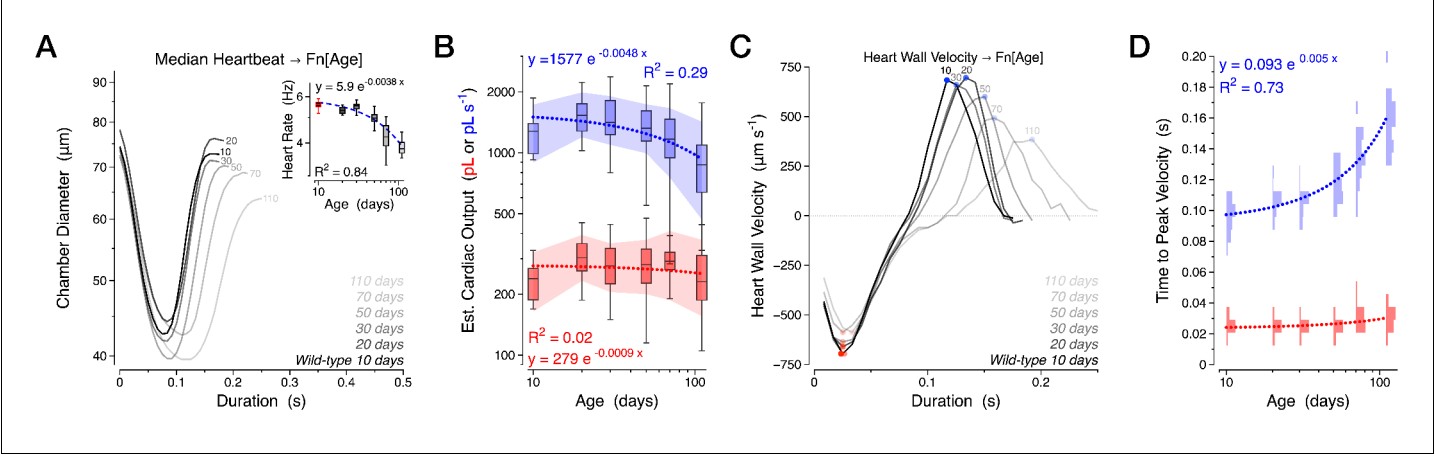

**Figure 2.** Normal aging is characterized by a diastolic decline with preserved contractility. Various cardiac functional parameters presented by age in a combined $w^{1118}$ and *Canton S* dataset, *n* = 18 to 30 animals per time-point: (**A**) Median heartbeat with heart rate (inset). (**B**) Estimated cardiac output per second (blue) and per stroke (red). (**C**) Median heart wall velocity with peak velocities of contraction (red dots) and relaxation (blue dots). (**D**) Probability histograms of the time from initiation of contraction to peak contraction velocity (red) and from the peak contraction velocity to peak relaxation velocity (blue). The shaded areas in panel B represent the mean +/− s.d., with regressions plotted as dotted lines.

The following source data and figure supplement are available for figure 2:

**Source data 1.** Median heartbeats for all individual animals in panel A.
**Figure supplement 1.** Further measures of normal aging.

Although we observed significant reductions in total cardiac output in aged animals (*Figure 2B*), the mechanical performance of each individual contraction was remarkably preserved. We observed no significant changes in stroke volume (*Figure 2B*, red), chamber diameter (*Figure 2—figure supplement 1C*), or fractional shortening (*Figure 2—figure supplement 1D*), suggesting that the observed decline in cardiac performance might reside in the temporal rather than spatial domain. Indeed, the most striking change associated with aging was an increased latency in transitioning from systole to diastole, which primarily reflected a decrease in the velocity of relaxation (positive values in *Figure 2C*) and an increase in the time period from peak contraction velocity to peak relaxation velocity (*Figure 2D*, blue, $R^2$ = 0.73). In contrast, the kinetics of contraction was much less affected (*Figure 2C,D*). Similar diastolic decline with preserved contractile performance accounts for an increasing fraction of heart failure cases in humans (*Borlaug, 2014*; *Sharma and Kass, 2014*), suggesting a conserved susceptibility to aging-related declines in the structural or biochemical processes facilitating relaxation.

## A pair of K2P subunits, *sandman* and *galene*, are required for terminating systole in aged animals

The rhythmic contraction and relaxation of the heart requires a precisely tuned series of ionic conductances that entrain the influx and efflux of calcium across the sarcolemnal and sarcoplasmic reticular membranes (*Bers, 2008*; *Monfredi et al., 2013*). In cardiomyocytes, outward potassium currents mediating repolarization are essential for terminating systole and suppressing dysrhythmic afterdepolarizations (*Nerbonne and Kass, 2005*). One physiological hallmark of failing hearts is the progressive loss of these repolarizing currents (*Beuckelmann et al., 1993*). We therefore investigated the repolarizing mechanisms maintaining diastolic function and normal rhythm in *Drosophila*. Previous work has implicated a number of potassium channels important in the repolarization to, or maintenance of, the cardiac resting potential in *Drosophila*, including KCNQ (*Ocorr et al., 2007*) and the K2P channel ORK1 (*Lalevée et al., 2006*).

While several voltage-gated potassium channels have well established roles in cardiac repolarization, a number of K2P channel family members are expressed in the human heart but their

physiological relevance remains an active area of investigation (*Schmitt et al., 2014*). TASK-1 ($K_{2P}3.1$) has been implicated in chronic atrial fibrillation, where TASK-1 protein levels are increased and action potential duration is shortened in a TASK-1 dependent manner relative to controls (*Schmidt et al., 2015*). In the larval heart of *Drosophila*, the K2P family member ORK1 appears to fine tune the rate of slow diastolic depolarization (*Lalevée et al., 2006*). Transcriptional profiling has previously revealed a putative two-pore potassium channel subunit, *CG8713*, recently named *sandman* (*Pimentel et al., 2016*), with enriched mRNA expression in the *Drosophila* heart relative to other tissues (*Robinson et al., 2013*). Reasoning that this gene may play a role in cardiac repolarization, we confirmed its expression in the fly heart using RT-PCR and generated a small deletion of *CG8713* and the adjacent gene *CG8712* (*Figure 3—figure supplement 1A,C*). At 50 days of age, the hearts of *sandman* mutants displayed a marked inability to transition from heart contraction (systole) to heart relaxation (diastole), with some animals rarely or never displaying heart relaxation (*Figure 3A*, *Video 2*).

This phenotype appears to be heart autonomous and specifically due to the loss of *sandman*, not *CG8712*. Expression of the *sandman* cDNA in cardiomyocytes using *R94C02::Gal4* rescued the functional defects observed, including the median heartbeat waveform (*Figure 3B*) and the probability map of fractional shortening versus systolic interval (*Figure 3C*). We next screened other putative K2P channel subunits and uncovered a similar heart-autonomous role the heart expressed K2P subunit *CG9194* (henceforth *galene*) (*Figure 3—figure supplement 1B,C*), for which knockdown also led to a marked decline in cardiac function. Heterozygous animals expressing dsRNA selectively targeting *galene* in the heart using *tinCΔ4::Gal4* displayed a dispersion in fractional shortening versus systolic interval (*Figure 3—figure supplement 2A*), a significant reduction in cardiac output per second and per stroke (*Figure 3—figure supplement 2B*), and a prolongation of the systolic interval relative to controls (*Figure 3—figure supplement 2C*). As in *sandman* mutants, this defect primarily reflects a difficulty in transitioning from systole to diastole; the median heartbeat waveform exhibited a clear reduction of diastolic function relative to controls (*Figure 3—figure supplement 2D*).

## The *sandman* phenotype displays an age-dependent progression

Defects in cardiac repolarization can manifest as life threatening arrhythmias, but their pathophysiology is complex due to compensatory repolarization reserve and arrhythmogenic electrical remodeling (*Nattel et al., 2007*). For example, atrial fibrillation exhibits an age-dependent penetrance that reflects self-reinforcing electrophysiological remodeling and normal aging. The action potential is shortened, which facilitates reentrant excitation, causing further shortening of the action potential thereby completing the feed-forward loop. Conversely, aging hearts exhibit action potential prolongation, which augments contractility but also predisposes the heart to Torsades des Pointes tachyarrhythmia, a ventricular rhythm defect that can lead to sudden cardiac death. Such age-dependent pathogenesis is evident in a *Drosophila* model of long QT syndrome where KCNQ potassium channel mutants develop progressive dysrhythmia (*Ocorr et al., 2007*).

We systematically quantified the age-dependent progression of diastolic failure in *sandman* mutants. *sandman* mutants displayed only minor defects in cardiac function early in life. Cardiac output per second and per stroke were statistically indistinguishable from wild-type at 10 days of age (*Figure 4A,B*), with only modest increases in the systolic interval (*Figure 4C*). However, by 30 days of age heart function was severely compromised; cardiac output per second and per stroke declined precipitously (*Figure 4A,B*) while the systolic and diastolic durations increased dramatically (*Figure 4C,D*). By 50 days of age, median total cardiac output per second had declined approximately 100-fold (*Figure 4A*). This progression can also be readily observed in two-dimensional probability maps of chamber diameter and fractional shortening (*Figure 4—figure supplement 1*) and can be rescued by expressing *sandman* cDNA selectively in cardiomyocytes of *sandman* mutants (*Figure 4A–D*, green boxplots) but not when expressed selectively in body wall muscles excluding the heart (see *Figure 4—figure supplements 2* and *3* for detailed statistics for all measurements by age and genotype).

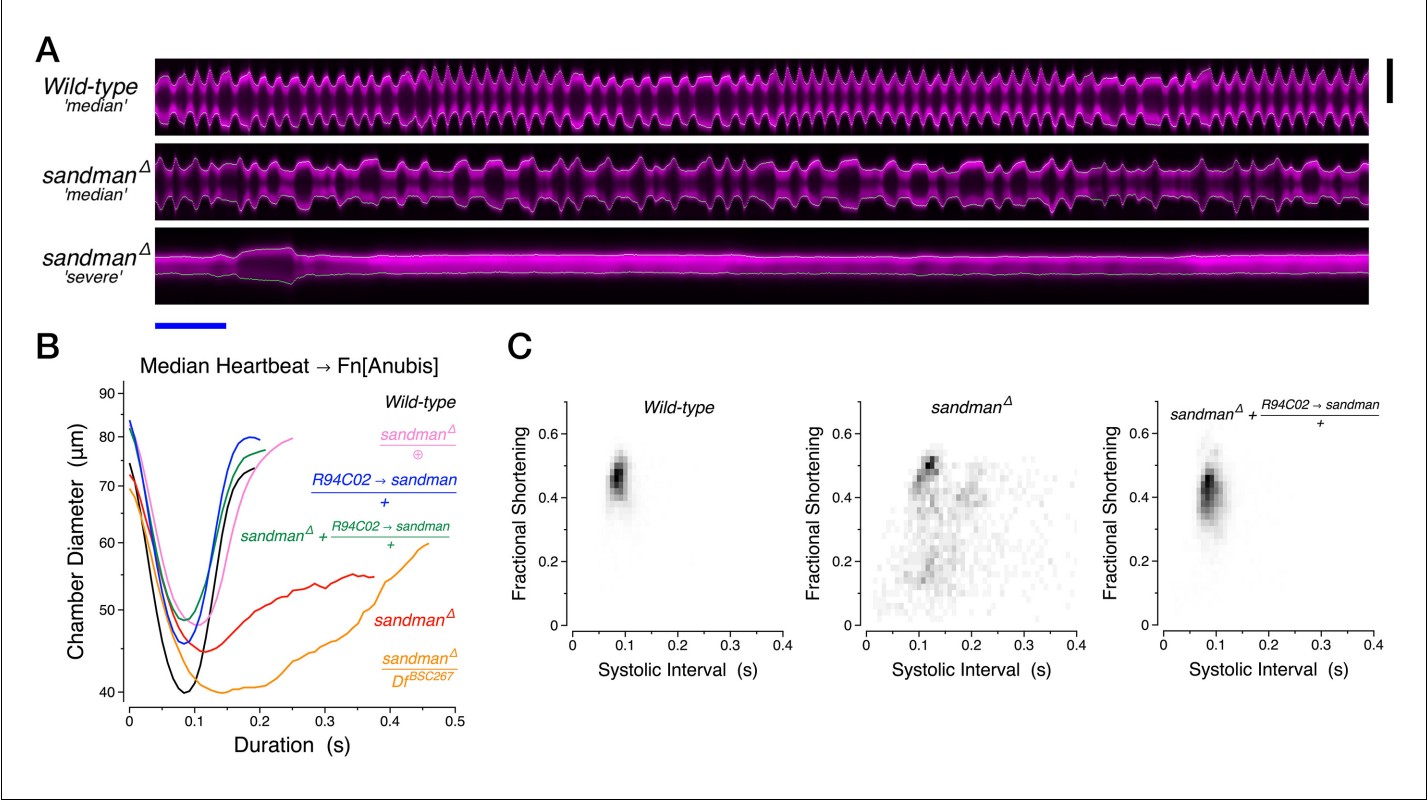

**Figure 3.** Diastolic failure in *sandman* mutants. (**A**) Representative YT kymographs of 50-day-old animals. Scale bars: (black vertical) 75 μm, (blue horizontal) 1 s. (**B**) Median heartbeat per genotype at 50 days of age. ⊕, clean excision of the mutagenic *piggyBac* insertion *e00867*. (**C**) Two-dimensional probability map of fractional shortening and systolic interval at 50 days of age. See also *Video 2*.

The following source data and figure supplements are available for figure 3:

**Source data 1.** Median heartbeats for all individual animals in panel B.

**Figure supplement 1.** *sandman* and *galene* genetic loci.

**Figure supplement 2.** RNAi knockdown of *galene* from birth through 40 days of age.

### *Sandman* and *galene* jointly encode a heteromeric potassium channel in vitro

Although historically considered background 'leak' currents that influence the resting membrane potential of cells, K2P channels are gated by diverse physiochemical stimuli and may therefore couple the activity of tissues to their environment (*Enyedi et al., 2010*). *sandman* was recently implicated as a dopamine-induced potassium current that operates as part of a homeostatic sleep switch (*Pimentel et al., 2016*). We characterized the electrophysiological behavior of *sandman* and *galene* by expressing them in Chinese hamster ovary (CHO) cells. Consistent with their similar phenotypes in vivo, *sandman* and *galene* most likely encode two subunits that form a heteromeric ion channel in vitro. Neither subunit expressed alone was sufficient to give rise to an appreciable conductance in CHO cells whereas co-expression resulted in large outward currents at depolarized potentials (*Figure 5A*). This conductance is highly selective for potassium and, in contrast to the open rectifier ORK1 (*Goldstein et al., 1996*), displays outward rectification that is only partially due to block by divalent cations (*Figure 5B*), a conserved feature of many K2P channel subtypes (*Schewe et al., 2016*). Importantly, this functional co-assembly appears selective; the two most closely related K2P subunits, CG1688 and CG10864, do not form functional heteromers with either Sandman or Galene (*Figure 5—figure supplement 1*). Heteromultimerization of potassium channel subunits is a well-

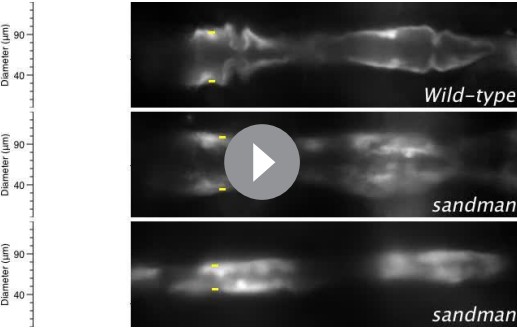

**Video 2.** 50-day *wild-type* and *sandman* heartbeat visualization. One-third speed videos of 50-day adult *wild-type* (upper video) and *sandman* (lower two videos) females, with heart wall position calls (yellow) and the attending transformation into heart chamber diameter as a function of time in a 1 s streaming window. The initiation and end of each contraction are specified by a red and blue triangle, respectively.

established mechanism for increasing electro-physiological diversity of tetrameric potassium channels as well as dimeric K2P channels (*Yang and Nerbonne, 2016*). Considering that several K2P channel subtypes are functionally silent as homodimers in vitro (*Enyedi et al., 2010*; *Goldstein et al., 2005*), heteromeric complementation, as observed between Sandman and Galene (*Figure 5A*), may be of considerable significance to the physiology of this family.

## The *sandman* phenotype likely reflects a loss of repolarization rather than structurally congestive remodeling

To our knowledge, *Drosophila* represents the only model organism where the pumping action of the heart is acutely dispensable for adult survival. We therefore characterized the terminal phenotype of *sandman* mutants to determine whether it may reflect congestive structural remodeling or a heart that is constitutively contracting. To differentiate these possibilities, we acutely injected the calcium chelator EGTA into the abdominal cavity of intact 110 days old *sandman* animals and compared the heart's contractile state before and immediately after. EGTA robustly terminated the persistent contractions observed in *sandman* mutants, establishing that cardiomyocytes were actively contracting in a $Ca^{2+}$ dependent manner rather than locked into congestive cytoskeletal remodeling (*Figure 6A*). Similarly, acute injections of the potassium ionophore valinomycin also terminated the prolonged contractions observed in *sandman* mutants (*Figure 6A*), confirming that the phenotype was dependent on the intracellular concentration of potassium ions and likely reflected a failure to fully repolarize.

To further characterize the constitutively contracted state of aged *sandman* hearts, we directly imaged sarcomere dynamics in vivo using a GFP protein trap of the z-line protein α-actinin (*Figure 6B*). We quantified the relative coherence of sarcomeres between left and right cardiomyocyte pairs by calculating the fraction of time in which they were moving in unison towards or away from the midline, which reflects systole and diastole respectively. As expected, wild-type sarcomeres contract and relax in unison (*Figure 6C*, *Video 3*). In *sandman* mutants, individual sarcomeres exhibited a fibrillatory state during extended systoles, contracting and relaxing out of phase with one another (*Figure 6D*, *Video 3*), suggesting that sufficient ATP was available locally to drive myosin activity, myosin dissociation and the various pumps that can sequester or extrude calcium. Because the contractile state of individual sarcomeres is intimately linked to local calcium cycling (*Bers, 2008*; *Hohendanner et al., 2013*; *Venetucci et al., 2008*), it is possible that the observed dyssynchrony of sarcomere dynamics reflects asynchronous local calcium rise and sequestration in the absence of a coherently cycling action potential.

## Discussion

### The *Drosophila* heart represents a reduced system for understanding fundamental mechanisms of heart function and aging

In this study, we establish a genetically accessible in vivo model for understanding the molecular mechanisms regulating cardiac performance during normal aging. Our imaging methodology permits the extended imaging of heart function in intact, unanesthetized animals for several hours, without measurable declines in cardiac performance (*Figure 1—figure supplement 1C,D*). The resolution and reproducibility of our measurements revealed specific changes to the heartbeat waveform as animals age. Using a total of 146 wild-type flies from six different ages, we visualized

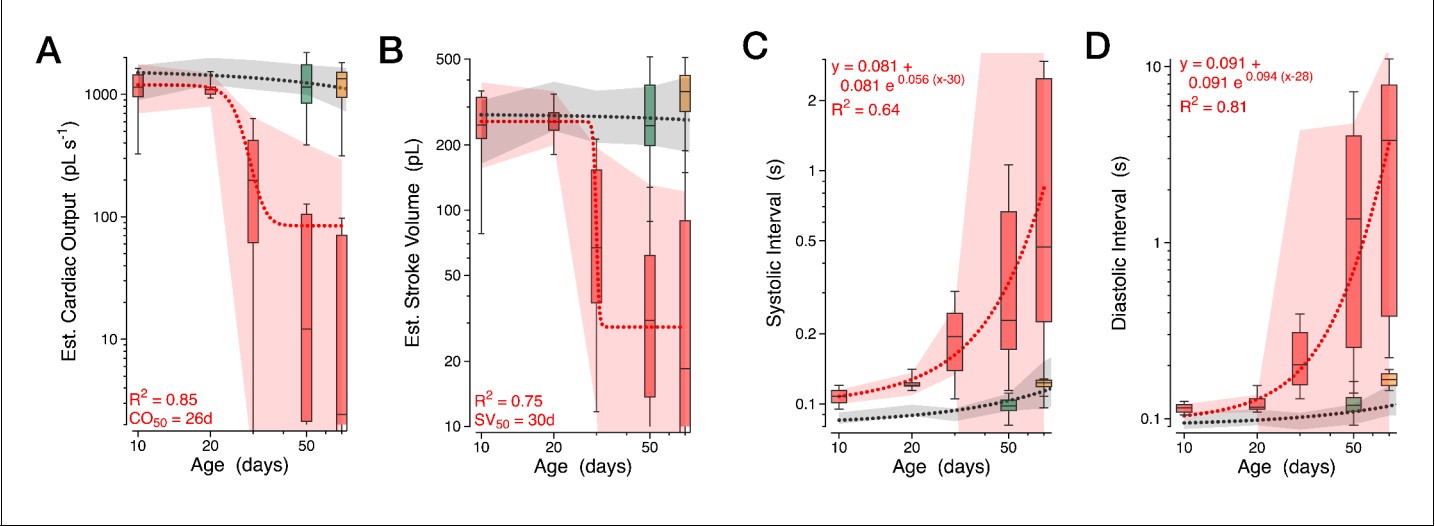

**Figure 4.** Progressive heart failure in *sandman* mutants. (**A–D**) Estimated cardiac output per second (**A**) and per stroke (**B**) were well fit by a Boltzmann sigmoidal regression and the systolic (**C**) and diastolic (**D**) intervals were well fit by single exponential growth regression curves for *wild-type* (grey), *sandman* (red), cardiomyocyte rescue of *sandman* using *R94C02::Gal4* (green) and the clean excision (pink) at specified ages. *n* = 7 to 27 animals per genotype and age. The shaded areas represent the mean +/− s.d., with the regressions plotted as dashed lines.

The following figure supplements are available for figure 4:

**Figure supplement 1.** Progressive loss of diastole in *sandman* mutants.

**Figure supplement 2.** Additional cardiac functional parameters for *sandman* mutants.

**Figure supplement 3.** Transgenic rescue in *sandman* mutants at 50 days of age.

approximately 65,000 heartbeats, reconstructing median heartbeat waveforms for each age (*Figure 2A*) and demonstrating a progressive decline in the kinetics of relaxation but not contraction (*Figure 2C,D*).

The throughput of this in vivo assay facilitates the identification of novel genes that establish and maintain cardiac function and can therefore complement more complex vertebrate models. In a small scale screen, we identified two K2P channel subunit genes, *sandman* and g*alene*, which together give rise to a heteromeric potassium channel that appears essential for terminating systole and promoting relaxation in aged animals. Our analysis of *sandman* mutants revealed that the pumping function of the heart is acutely dispensable for adult *Drosophila* survival under laboratory conditions (*Figure 3A*, *Video 2*). Although cardiac function is not required for zebrafish embryogenesis or the first week of development, heart function is otherwise essential for the survival of all adult vertebrates (*Staudt and Stainier, 2012*). Differences in heart dispensability likely originate in the decoupling of gas exchange and heart function in larval zebrafish or insects, where sufficient gas exchange can occur via local diffusion. This opens a unique window for understanding the physiological transition to, and maintenance of, fibrillatory arrest without the confounds of organismal or cardiomyocyte death. The future development of in vivo approaches for monitoring cytosolic and sarcoplasmic reticular $Ca^{2+}$ levels in the beating heart will facilitate a clearer mechanistic view of the biophysical mechanisms initiating and sustaining fibrillatory arrest.

## Sandman is required to maintain diverse biological oscillators

Oscillatory behaviors exist across diverse timescales, from migratory cycles in birds to ultrafast spiking neurons in the auditory system. One conserved feature of oscillators is their capacity to be modulated by internal and external cues, thus adapting the system to immediate physiological needs or entraining the phase of the cycle to the external world. Interestingly, *sandman* appears to play a central role in two radically different biological oscillations: organismal sleep-wakefulness

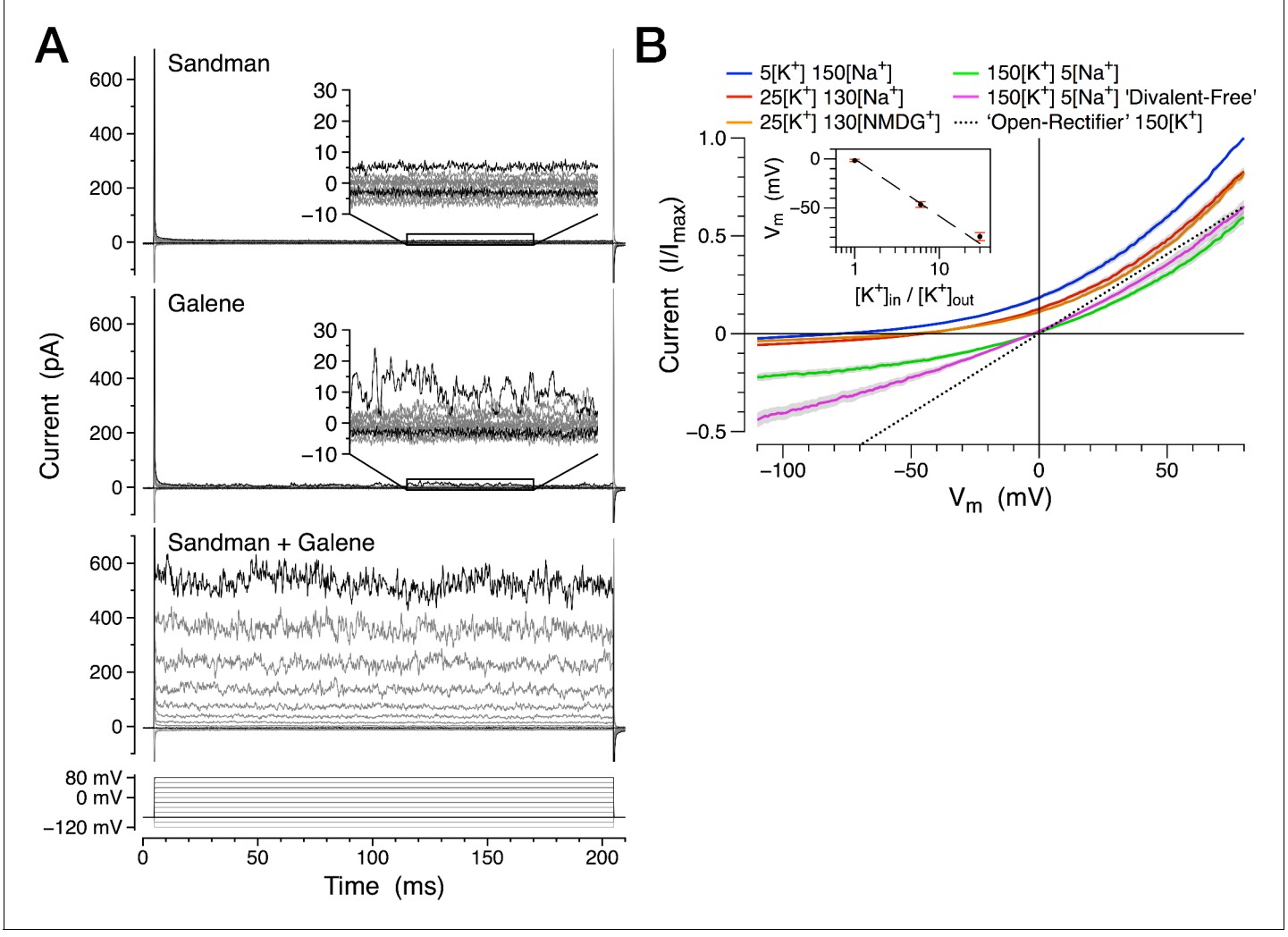

**Figure 5.** *sandman* and *galene* jointly encode a potassium channel. (**A**) Representative whole-cell currents in physiological $K^+$ and $Na^+$ gradients from Sandman (*n* = 5), Galene (*n* = 6), and co-transfection of both (*n* = 11) during voltage steps (below). (**B**) Normalized whole-cell currents from voltage ramps in various bath solutions. The dotted line plots the I/V curve for a hypothetical ion channel with no rectification in symmetric $K^+$. The inset plots the observed reversal potential compared to a potassium-selective conductance (dashed line) at various $[K^+]_{in/out}$ ratios. The internal pipet solution is (in mM) 150 $K^+$, 5 $Na^+$, 3 $Mg^{2+}$, 161 $Cl^-$, 10 HEPES, pH 7.4 (in mM). The bath solution $[K^+]$ and $[Na^+]$ or $[NMDG^+]$ are as indicated (in mM), excepting the 'Divalent-free' solution which substitutes 2 mM EDTA for the divalent cations. *n* = 9 cells. All pooled data represent the mean +/− s.d. All voltage potentials are relative to ground.

The following source data and figure supplement are available for figure 5:

**Source data 1.** Normalized current-voltage data for panel B.

**Figure supplement 1.** Sandman and Galene do not form functional heteromeric channels with the closely related K2P subunits CG1688 or CG10864.

(*Pimentel et al., 2016*) and the contraction-relaxation of aging hearts (this paper). Pimental and colleagues demonstrated that Sandman-dependent potassium currents are upregulated by dopamine via a G-protein cascade that is pertussis toxin sensitive. This increase in potassium 'leak' significantly reduces the excitability of sleep-promoting dFB neurons and therefore tips the balance of the cycle towards wakefulness. Phenotypically, our results suggest that *sandman* plays a critical role in maintaining a contraction-relaxation oscillator during aging. Loss of *sandman* does not grossly perturb cardiac rhythmicity in young animals but the balance between contraction and relaxation degenerates catastrophically with age, leaving the heart in a persistently contracted state.

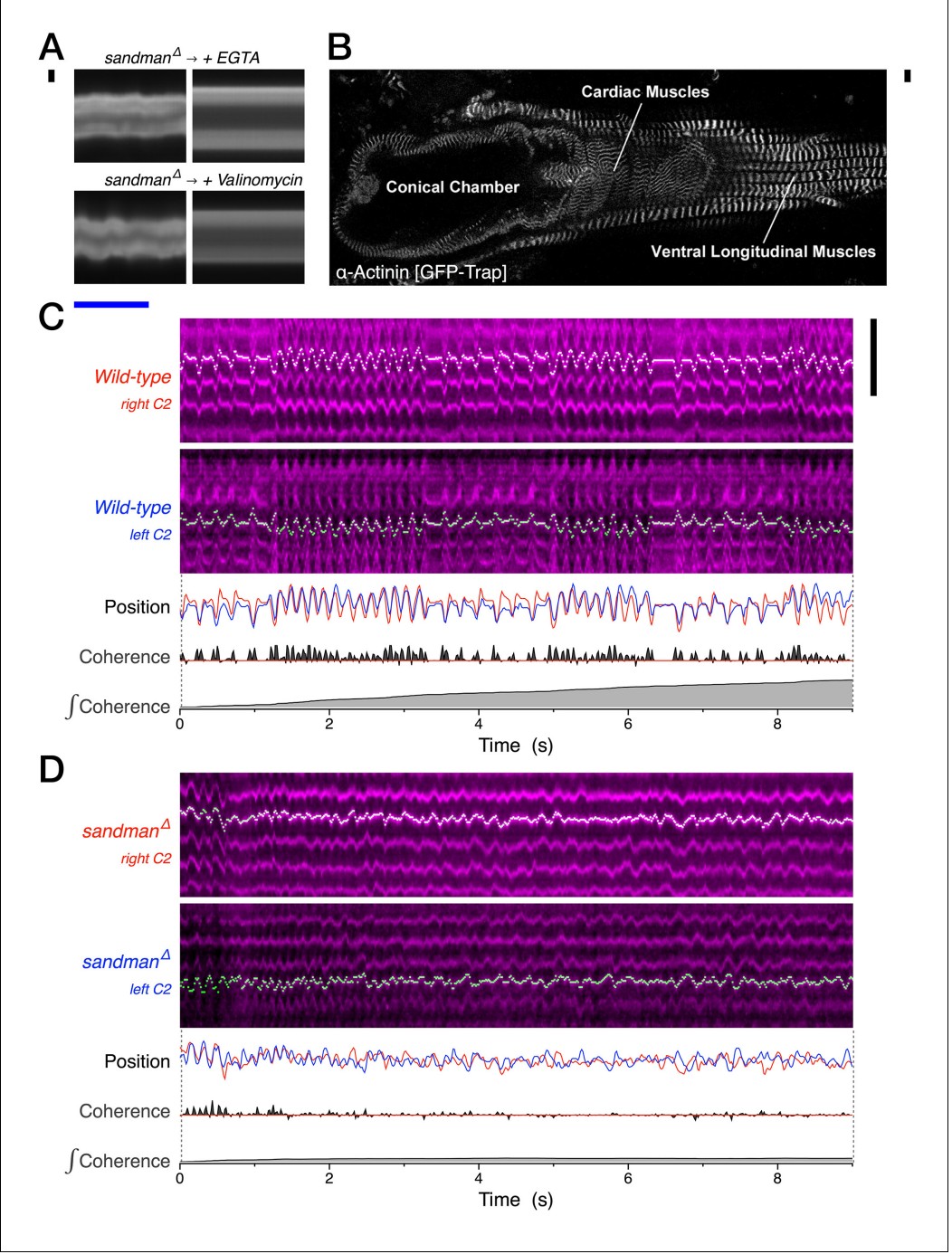

**Figure 6.** In vivo pharmacology and sarcomere dynamics implicate dyssynchronous and regenerative Ca²⁺ in maintaining persistent systole. (**A**) Representative heart kymographs from 110-day-old *sandman* males before and acutely after intra-abdominal injection of the Ca²⁺ chelator EGTA or the potassium ionophore valinomycin. *n* = 3. (**B**) Micrograph of a dissected adult *Drosophila* expressing a GFP trap of the z-line protein α-*actinin* [CC01961]. (**C–D**), Representative kymographs (magenta) of second chamber right and left cardiomyocyte sarcomere dynamics from intact 30-day-old animals, as visualized intravitally using the α-*actinin* GFP-trap. Automated detection of one z-line for each cardiomyocyte (green/white), quantified as relative position over time (upper trace, left cardiomyocyte signal inverted), with net coherence between z-lines (middle trace) and integral coherence (bottom trace). Scale bars: (black vertical) 10 μm, (blue horizontal) 1 s. *n* = 9 for *wild-type* and four for *sandman*. See also *Video 3*.

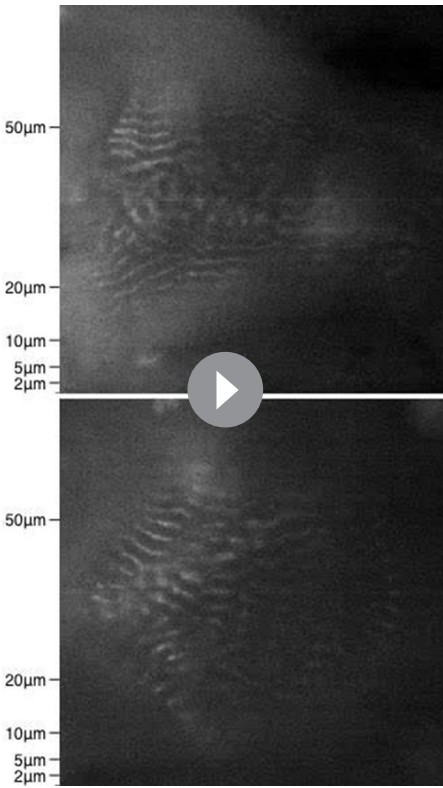

**Video 3.** 30 day *wild-type* and *sandman* sarcomere dynamics. Real-time videos of 30-day adult *wild-type* (upper video) and *sandman* (lower video) females, visualizing the z-lines of the dorsal aspect of the cardiomyocyte pair just posterior to the ostial valves of the second chamber using a protein trap of α-*actinin* [CC01961].

While the molecular mechanism by which pertussis-toxin sensitive G-protein signaling upregulates Sandman currents is not yet understood, a variety of downstream signaling events are known to affect the activity and localization of ion channels (*Inanobe and Kurachi, 2014*). Similar pathways play a significant and complex role in the neurohormonal regulation of the mammalian heart, for instance by the counterbalanced G-protein coupled pathways activated by the sympathetic and parasympathetic nervous systems (*Mangoni and Nargeot, 2008*). Aging human hearts exhibit diminished responsiveness to these modulatory pathways, suggesting senescent defects in signal transduction or a system driven to the limits of its dynamic range (*Kaye and Esler, 2008*). Similarly, work in *Drosophila* has demonstrated that the heart is sensitive to neural input (*Dulcis and Levine, 2005*). It will be interesting to determine whether extrinsic or intrinsic regulatory signals tune Sandman activity during aging and thus optimize the heart's contraction-relaxation balance in a fashion analogous to that observed in the sleep center of the brain.

## Diastolic decline with preserved contractility may represent a conserved feature of aging cardiomyocytes

Young hearts possess abundant repolarizing reserve and can often compensate for losses in individual conductances without impeding the heart's ability to rapidly relax (*Nattel et al., 2007*). For reasons that are not well understood, this reserve steadily declines with age in humans and animal models, leaving the heart susceptible to late-onset dysrhythmias. The cardiac phenotype observed upon loss of *sandman* is strikingly age-dependent, transitioning from grossly normal heartbeat in young animals to complete fibrillatory arrest later in life. Our in vivo pharmacological results implicate defects in action potential repolarization, consistent with the rectification behavior of the Sandman/Galene heteromeric channel (*Figure 5B*). These observations suggest that a progressive loss of repolarization reserve may represent a conserved feature of cardiac aging. However, the pathophysiological mechanisms underlying this age-dependent loss are not known.

In humans, an increasing number of heart failure cases display significant defects in relaxation while largely preserving contractile performance, termed 'Heart Failure with preserved Ejection Fraction' (HFpEF) (*Borlaug, 2014*; *Sharma and Kass, 2014*). Owing to the closed nature of the vertebrate cardiovascular system, the pathophysiology of such diastolic decline reflects a diverse and complicated assemblage of causal mechanisms including cardiomyocyte dysfunction, structural remodeling and increased vascular resistance. Despite dramatic differences in the architecture of their cardiovascular systems, aging *Drosophila* similarly exhibit preferential diastolic decline (*Figure 2*). Although many pathways likely contribute to the pathophysiology of HFpEF, our results suggest that cardiomyocytes across phylogeny may possess a conserved active mechanism driving this differential outcome.

Previous work has uncovered several potential mechanisms that may differentially regulate systolic and diastolic tone. Mechanically, a growing body of literature has implicated isoform switching and postranslational modifications of the giant macromolecular spring Titin (*Linke and Hamdani, 2014*). Patients experiencing heart failure with preserved ejection fraction exhibit Titin

hypophosphorylation, which increases the resting tension of cardiomyocytes and impairs diastolic function (*Hamdani et al., 2013*). Conversely, increasing Titin compliance experimentally has a beneficial effect on diastolic performance but compromises elements of systolic function, notably the Frank-Starling reflex (*Methawasin et al., 2014*). Electrophysiologically, there is considerable evidence that action potential duration is differentially remodeled in aging hearts, heart failure and atrial fibrillation (*Beuckelmann et al., 1993*; *Heijman et al., 2014*; *Janczewski et al., 2002*), which may ameliorate the initial dysfunction but predisposes the heart to later dysrhythmia. In animal models and patients exhibiting chronic heart failure, several potassium channel subunits appear downregulated (*Beuckelmann et al., 1993*; *Nattel et al., 2007*), prolonging the action potential in an apparent attempt to augment contractile function. Such decline has also been observed in aging *Drosophila*, where KCNQ transcripts appear downregulated and animals lacking KCNQ display progressive dysrhythmia in situ (*Ocorr et al., 2007*).

Together, these observations suggest that contractile function may be adaptively regulated by mechanisms that are conserved across phylogeny and that age-related diastolic decline and increased susceptibility to dysrhythmia may represent unintended side effects of this compensation. Our results also implicate a heteromeric potassium channel as a critical effector for maintaining normal rhythmicity during aging, suggesting that age-dependent cardiomyocyte membrane properties may play a key role in maintaining cardiac function into old age. Several important questions remain unaddressed. How does the heart or the brain sense, integrate and respond to alterations in contractile efficacy? Do these mechanisms exacerbate declines in diastolic function during normal aging and to what extent do these homeostatic mechanisms contribute to the pathogenesis of heart disease? The acute dispensability of the fly heart pumping, the extensive genetic tools available in *Drosophila*, and the intravital imaging system we developed provide an exciting opportunity for exploring this dynamic nexus between cardiac physiology, aging and disease.

## Materials and methods

### DNA constructs

DNA plasmids used for fly transgenesis and heterologous cell transfection were assembled using standard molecular biological techniques and sequenced to confirm accuracy and identity. Sub-cloning details and plasmid descriptions are presented in *Table 1*.

### RT-PCR

Fifty fly hearts were microdissected from the adult abdomen of mixed age $w^{1118}$ flies and total RNA was isolated using the ZR RNA MicroPrep Kit (Zymo Research, Irvine, CA). cDNA transcripts were generated using Superscript III RT and oligo dT primers (Thermo Fisher) and PCR amplification was tested using GoTaq Green Master Mix (Promega) using the following primers:
*sandman* 5' TACAGAGCGCGCAAACATA 3' AGGATTTCCGGCTACCTATCG
*galene* 5' TTTGTGGCTCGTACGGATCG 3' CTAATTTGCCGCTCGGTTGG

### *Drosophila* genetics and husbandry

All chromosomal aberrations and transgenic insertions used in this study are detailed in *Table 2*. Transgenic elements generated in the course of this study were inserted into specific *attP* docking sites within the *Drosophila* genome using *phiC31*-mediated integration (*Bischof et al., 2007*). The deletion of *sandman*, e00867-e00152, was generated using FLP-FRT mediated recombination of two *piggyBac* elements in trans and confirmed using PCR (*Parks et al., 2004*; *Thibault et al., 2004*). Clean excision of the *piggyBac* elements was performed as previously described (*Parks et al., 2004*; *Thibault et al., 2004*). Animals were raised on standard cornmeal, yeast, agar, molasses formula and kept in a diurnal 12 hr light: 12 hr dark 70% humidity-controlled incubator (Darwin Chambers, St. Louis, MO). All experimental adult flies were raised at 20°C excepting RNAi knockdown, which was performed at 25°C to increase dsRNA expression. For aging, 20 female and 10 male flies were transferred to fresh unyeasted vials every 7 days. Approximately 18 hr before imaging, a wedge of rinsed and water saturated cellulose acetate (Genesee Scientific, San Diego, CA) was added to the fly vials to ensure adequate hydration of the flies. CG9194 was named *galene* after the ancient Greek goddess of calm seas.

**Table 1.** DNA constructs.

| Plasmid ID | Plasmid name | Insert source | 5' Primer | 3' Primer | Destination Vector | Restriction Subcloning | Comments |
|---|---|---|---|---|---|---|---|
| pMK1 | 10xUAS-IVS-Syn21-tdTomato-p10 | pDEST HemmarR (Addgene #31222) | ataaggtaccAACTTAAAAAAAAAAA TCAAAATGGTGAGCAAGGGCGAG | atatttctagaTTACTTGTACAGCTCG TCCATGCC | pJFRC81 (Addgene #36432) | KpnI - XbaI | Intermediate plasmid - for fly transgenesis (Han et al., 2011; Pfeiffer et al., 2012) |
| pMK3 | R94C02::tdTomato | Janelia Farms, amplified from Drosophila genome | tactagtACTTTTCCGCGCCCGTCTG | atatgctagcGGAAACAGACGCAAAGAC TGAC | pMK1 (this paper) | HindIII - NheI | Cardiomyocyte enhancer expressing tdTomato for fly transgenesis - 5' primer was phosphorylated to facilitate blunt ligation using Klenow fragment |
| pMK17 | pAc5.1B_GFP-CG8713 | BDGP cDNA RE21922 in clone #UFO03925 | aaaagcggccgcATGTCCTCCCGACGC | atatttctagaTTAGGAGGTGCGGCAC | pAc5.1B_GFP (Addgene #21181) | NotI - XbaI | Intermediate plasmid - for S2 cell expression (Stapleton et al., 2002) |
| pMK18 | pAc5.1B_GFP-CG10864 | PCR from Drosophila genome (single exon) | aaaagcggccgcATGGCCAGCAAA TTTCAGAG | atatttctagaCTAGTAGTAATCATCCTCG TAC | pAc5.1B_GFP (Addgene #21181) | NotI - XbaI | Intermediate plasmid - for S2 cell expression |
| pMK21 | pAc5.1B_GFP-CG1688 | BDGP cDNA GH04802 in clone #UFO05944 | aaaagcggccgcATGTCCGACG TTGAGCAG | atatttctagaTTATCCATCCGCGCGGG | pAc5.1B_GFP (Addgene #21181) | NotI - XbaI | Intermediate plasmid - for S2 cell expression |
| pMK22 | pACU_GFP-CG8713 | pAc5.1B_GFP-CG8713 (this paper) | KpnI - ApaI insert from source vector | KpnI - ApaI insert from source vector | pACU (Addgene #58373) | KpnI - ApaI | UAS::GFP-CG8713 rescue construct - for fly transgenesis |
| pMK23 | peGFP_C1-CG8713 | pAc5.1B_GFP-CG8713 (this paper) | EcoRI - ApaI insert from source vector | EcoRI - ApaI insert from source vector | peGFP_C1 (Clontech) | EcoRI - ApaI | For expression in mammalian heterologous cells |
| pMK24 | pAC5.1B_eGFP-CG9194 | BDGP cDNA FI03418 in clone #UFO11253 | aaaagcggccgcATGTCGGG TAGGCGGGGCCCA | gcagcctctagaCTAATCCTCATCCTGC TCGTCGTCATCATCC | pAc5.1B_GFP (Addgene #21181) | NotI - XbaI | Intermediate plasmid - for S2 cell expression |
| pMK25 | peGFP_C1-CG9194 | pAC5.1B_eGFP-CG9194 (this paper) | EcoRI - SacII insert from source vector | EcoRI - SacII insert from source vector | peGFP_C1 (Clontech) | EcoR1 - SacII | For expression in mammalian heterologous cells |
| pMK29 | peGFP_C1-CG1688 | pAC5.1B_eGFP-CG1688 (this paper) | tataGCTAGCggtaccaacatggtgagcaagg | tatacgggccctctagaTTATCCATCC | peGFP_C1 (Clontech) | NheI - ApaI | For expression in mammalian heterologous cells |
| pMK30 | peGFP_C1-CG10864 | pAC5.1B_eGFP-CG10864 (this paper) | EcoRI - ApaI insert from source vector | EcoRI - ApaI insert from source vector | peGFP_C1 (Clontech) | EcoR1 - ApaI | For expression in mammalian heterologous cells |

**Table 2.** *Drosophila* genomic aberrations and transgenic insertions.

| Chromosomal Element | Location | Source | Description |
|---|---|---|---|
| *R94C02::tdTomato* | Chr. II (attP40) | This paper | R94C02 enhancer expressing tdTomato in cardiomyocytes and a subset of other muscles within the adult fly. Integrated into the attP40 docking site. We used this transgenic for all heart wall imaging experiments. |
| *PBac{RB}CG8713* [*e00867*] | Chr. II | Exelixis via Bloomington | piggyBac{RB} insertion proximal to the 1st splice acceptor of the *CG8713* mRNA. This insertion has a strong heart function phenotype. |
| *PBac{RB}CG8712* [*e00152*] | Chr. II | Exelixis via Bloomington | piggyBac{RB} insertion into the *CG8712* coding sequence. This insertion does not have a discernible heart phenotype. |
| *PBac{RB}CG8713_CG8712* [*e00867_e00152 deletion, mW+*] | Chr. II | This paper | FLP-FRT mediated deletion of the genomic DNA between *e00867* and *e00152*, which comprises the entire *CG8713* coding sequence and most of the *CG8712* coding sequence. RB(+) to RB(+) recombination reconstitutes a single PBac{RB} after the intervening sequence is deleted. |
| [*e00867_e00152 deletion*] | Chr. II | This paper | Clean excision of *PBac{RB}CG8713_CG8712* [*e00867_e00152 deletion, mW+*]. This chromosome harbors the deletion without any residual pBac[RB] sequence. |
| *CyO, P{Tub-PBac\T}* | Chr. II | Bloomington | Source of piggyBac transposase activity for generating clean excisions, which were identified by the loss of mini-white eye pigmentation. |
| *e00867* [*clean excision*] | Chr. II | This paper | Clean excision of *PBac{RB}CG8713* [*e00867*] which reverts the observed heart phenotype. |
| *UAS::GFP-CG8713* | Chr. III (vk00005) | This paper | nsertion of plasmid pMK22 into the attP docking site *vk00005*. Used to demonstrate cardiomyocyte rescue of *CG8713* (*anubis*). |
| *R94C02::Gal4* | Chr. III (attP2) | Bloomington | In a screen for new heart-specific Gal4s, we found *R94C02::Gal4* as a complement to *tin.CΔ4::Gal4*. It is expressed in cardiomyocytes and a subset of other muscles. (Pfeiffer et al., 2008) |
| *tin.CΔ4::Gal4* | Chr. III | Manfred Frasch | *tin.CΔ4* is expressed in the heart and a subset of other muscles within the fly. (Lo and Frasch, 2001) |
| *R37B05::Gal4* | Chr. III (attP2) | Bloomington | In a screen for new heart-specific Gal4s, we found *R37B05::Gal4* to be expressed in bodywall muslces but not the heart. |
| *Df(2R)BSC267* | Chr. II | Bloomington | A molecularly defined deletion spanning the *CG8713* (*sandman*) locus. |
| *Df(3L)BSC431* | Chr. III | Bloomington | A molecularly defined deletion spanning the *CG9194* (*galene*) locus. |
| *P{KK110628}VIE-260B* | Chr. II | VDRC #v108758 | UAS::CG9194 dsRNA for tissue-specific knockdown of *galene*. UP-TORR does not predict any off-target effects. |
| *P{w[+mC]=PTT-GC Actn* [*CC01961*] | Chr. X | Bloomington, this paper | GFP translational fusion of α-Actinin, which is localized to the z-lines of all muscles. We recombined away the *yellow* and *white* alleles. |
| *Canton S* | N/A | Jan Lab | Wild-type stock |
| *w* [*1118*] | Chr. X | Jan Lab | Wild-type stock, outcrossed to Canton S 6x |
| *M{vas-int.Dm}ZH-2A; +; PBac{y[+]-attP-9A}VK00005* | Chr. X and III | Bloomington | Used for phiC31 mediated integration. |
| *P{nos-phiC31\int.NLS}X; P{CaryP} attP40* | Chr. X and II | Bloomington | Used for phiC31 mediated integration. |

## Intravital imaging

We used a modified Olympus BX51WI microscope for all video acquisition. Excitation light was provided by a DMD projector (DS + 6K-M, Christie Digital Systems, Cypress, CA) with the green light excitation intensity (538–568 nm, 4.3 mW/mm$^2$) and spatial extent controlled by PsychoPy (*Peirce, 2007*), using a standard 8 bit RGB tiff file as the signal. The projector was coupled to the microscope using a relay formed by two 150 mm focal length achromatic doublet lenses (Thorlabs Inc., Newton, NJ), placed into a tube assembly and attached to a second camera port above the vertical illuminator (Olympus U-DP and U-DP1XC), with the filter cube installed in the U-DP. We utilized a 20x/1.0 NA water-immersion objective (Olympus) and a sCMOS camera (PCO-Tech Inc., Romulus, MI), triggered using the vertical-sync of the projector signal as frequency doubled by a Master-8 stimulator (AMPI Jerusalem, Israel). Each fly was briefly anesthetized using ice, coupled to a No. one coverslip using Norland Optical adhesive #61 cured using a 365 nm 3 watt UV LED (LED Engin Knc., San Jose, CA) for 10 s and allowed to recover for 10 min before imaging. The coverslip was mounted to the microscope using an assembly containing a small goniometer (Thorlabs GN-05) which allowed

the pitch of each fly to be optimized. The optical path is as follows: water immersion objective, coverslip, optical cement, and intact fly. Images were acquired in global shutter mode using a 255 MHz clock with a 6 ms exposure at 120 frames per second. Each animal was recorded for 90 s, with 10,800 frames written directly to an array of 15 spindle disks using a RAID controller with write back cache enabled (LSI SAS9280, Avago Technologies, San Jose, CA). Acquisitions were Gaussian downsampled to 1 µm per pixel and converted to 8 bit using an ImageJ script prior to analysis. All heart wall videos in the manuscript utilized the same transgenic heart marker, *R94C02::tdTomato*[*attP40*], as heterozygotes.

### Heartbeat digitization

Using a manually-assisted ImageJ script, YT orthogonal kymographs of the heart wall just anterior to the ostial valve of the second heart chamber were generated. At this time, we also manually traced the systolic lumenal area of the second heart chamber, as specified by an average projection of the entire 10,800 frame stack. This assisted script calculated the area of the second chamber in systole and its length, which were appended to the image as metadata for later use. A second ImageJ script detected the right and left heart wall positions in the kymograph independently using a maximal contrast algorithm, refined by a four pixel maximum intensity search medial to that call. These calls were subsequently low pass filtered using an 8-pole 12 Hz Bessel filter (filter poles calculated using online software at http://www-users.cs.york.ac.uk/~fisher/mkfilter/trad.html). Rapid transient displacements resulting from errors in the heart wall detection algorithm were recursively smoothened using preceding and subsequent heart wall position calls. Analysis for the knockdown of *galene* was performed single-blind, but all other experiments were not blinded to genotype.

### Heartbeat segmentation

We automatically segmented each heartbeat into discrete contraction and relaxation events and analyzed these events in detail using VisualBasic scripts written for Diadem 2011 (National Instruments, Austin, TX). The following represents a summary of the material algorithms used to generate the functional parameters presented herein. First, we detected all contraction and relaxation events that fulfilled minimum velocity (75 µm/s), duration (24 ms) and displacement (2.5 µm) criteria. We then eliminated all prospective events where the two heart walls were not moving in unison, which resolved the residual fraction of false heart wall position calls. We next eliminated excess contraction and relaxation calls using nested timing and amplitude criteria so that the principal contractions and relaxations are interleaved 1:1. The initiation and end of contractions were then calculated by walking back to the last zero velocity time point before each contraction and relaxation. The software then acquired the heart diameters when contraction initiated and ended, which represent the end diastolic diameter (EDD) and end systolic diameter (ESD), respectively. These heartbeats were subsequently refined by consolidating compound contractions or relaxations not separated by a minimum duration (24 ms), below the minimum fractional shortening (0.04) or not exhibiting sufficient coherence between the two heart walls.

### Heartbeat analysis

A variety of functional parameters were calculated from the segmented heartbeat waveforms and are italicized and underlined for ease of reference. The *end diastolic diameter (EDD)* and *end systolic diameter (ESD)* were calculated as described in the previous section. We also quantified *heart chamber diameter* across the cardiac cycle, denoted by $\forall$. The *systolic interval* is the duration of contraction. The *diastolic interval* is the time elapsed between the end of contraction and the initiation of the next contraction. We observe only the occasional pause at the end diastolic diameter in *Drosophila*. The *fractional shortening* is the percentage change in diameter for each contraction where FS = (EDD - ESD)/EDD. The *stroke volume* of each heartbeat was estimated by modeling the heart as a radially contracting cylinder, using the average systolic diameter and fixed chamber length, in microns, generated for each animal during digitization as fixed variables. The average chamber systolic diameter was divided by the mean ESD to scale the measured ESDs onto the heart model. The estimated chamber systolic diameter for each heartbeat is therefore the ESD times this scaling factor. The estimated chamber diastolic diameter is then calculated by dividing the estimated chamber systolic diameter by the fractional shortening. The stroke volume in picoliters is therefore:

$$SV = \frac{\pi \cdot chamberLength \cdot chamberDiastolicRadius^2}{1000} - \frac{\pi \cdot chamberLength \cdot chamberSystolicRadius^2}{1000}$$

*Cardiac output per second* was calculated by integrating all stroke volumes that occurred in each second. The *heart rate* is the inverse of the duration between the initiation of adjacent contractions while *contractions per second* is the integral number of contractions the occurred in each second. The *median heartbeat* was derived by extracting the median diameter at each time point for time-aligned heartbeats until the median duration was reached. Quartiles were calculated similarly. Longer duration and more weakly relaxing heartbeats attenuate the apparent EDD at the end of the median heartbeat which is why a diameter mismatch between the initiation and end of the median heartbeat develops in animals exhibiting diastolic dysfunction. The *heart wall velocity* for each dataset was calculated using the derivative of the median heartbeat for that dataset. The *time to peak velocity* of contraction, and from peak contraction to peak relaxation, was calculated for each animal using their median heart wall velocities.

## Heartbeat visualization

We generated all graphs using DataGraph 4$\beta$ (Visual Data Tools) using the data output from Diadem 2011. Two-dimensional probability maps of chamber diameter and event duration were calculated in Diadem 2011 by aligning all contractions and heartbeats to the initiation of contraction and then displaying the normalized probability in DataGraph 4$\beta$. Probability maps of fractional shortening and systolic interval were similarly normalized to the maximum observed probability. Because *sandman* mutants displayed considerable phenotypic variability from animal to animal (see *Figure 3A* and *Video 2*), we did not find all datasets to exhibit normal distribution. Therefore, nearly all data is shown using Tukey boxplots so that the distributions can be accurately compared. The exception to this was when stroke volumes were juxtaposed with cardiac output per second; we presented stroke volumes as mean +/− s.d. for ease of visualization. Lastly, representative images of the heart, kymographs, and videos were overlaid with the heartbeat digitizations using ImageJ scripts.

## Statistical analysis

To more accurately reflect sample size and variability, we considered individual heartbeats as interdependent and performed all statistics using the mean values for each animal. The letter *n* therefore denotes the number of independent biological experiments, detailed in *Table 3*. Pilot experiments established that the principal phenotypes and their rescue were sufficiently robust to be statistically tested using moderate sample sizes. Because not all datasets exhibited normal distributions, we utilized a non-parametric statistical test, Kruskal-Wallis followed by Dunn's Multiple Comparison Test. ANOVA followed by Tukey's multiple comparison test reported similar results without material deviations in significance. For the sensitivity tests to excitation light intensity and mounting duration presented in *Figure 1—figure supplement 1*, we utilized paired one-way ANOVA followed by Holm-Sidak's multiple comparisons test. In all panels, significance is represented as ns = not significant, */#$p<0.05$, **/##$p<0.01$, ***/###$p<0.001$. Statistical tests were performed in Prism 6 software (Graph-Pad, La Jolla, CA). Fits were generated in DataGraph 4$\beta$ using a linear least squares minimization method of the animal means, with outliers exceeding one and one-half the standard deviation excluded from the regression analysis.

## Electrophysiology

Patch pipettes with resistances of 3.5–5 M$\Omega$ were pulled from borosilicate glass capillaries (1.5 mm O.D., 0.86 mm I.D., Sutter Instruments, Novato, CA) using a P-1000 pipette puller (Sutter Instruments) and fire-polished using a microforge (MF-830, Narishige, Tokyo, Japan). Authenticated and mycoplama tested Chinese hamster ovary cells (CHO-K1), acquired from the University of California San Francisco Core Facility via the European Collection of Authenticated Cell Cultures (ECACC 85051005), were grown in F12-K media supplemented with 10% fetal bovine serum and passaged fewer than nine times. 70% confluent 30 mm petri dishes were transfected overnight with FuGene 6 (Promega, Madison, WI) using 1 µg total plasmid. The cells were then replated onto poly-L lysine coated coverslips and allowed to recover for two hours before recording. Pipettes were mounted

**Table 3.** Intravital imaging sample sizes.

| Genotype and age | *n* | Genotype and age | *n* |
|---|---|---|---|
| w1118 10 days | 10 | w1118 + R94C02 > sandman cDNA 50 days | 12 |
| w1118 20 days | 12 | sandman∆ + R94C02 > sandman cDNA 50 days | 27 |
| w1118 30 days | 13 | sandman∆ + tinC∆4 > sandman cDNA 50 days | 12 |
| w1118 50 days | 17 | sandman∆ + R37B05 > sandman cDNA 50 days | 13 |
| w1118 70 days | 12 | sandman∆ / BSC267[Df] 50 days | 12 |
| w1118 110 days | 13 | sandman∆ / clean excision 50 days | 13 |
| Canton S 10 days | 8 | sandman∆ / BSC267[Df] 70 days | 15 |
| Canton S 20 days | 11 | sandman∆ / clean excision 70 days | 7 |
| Canton S 30 days | 14 | tinC∆4 control 40 days 25C | 15 |
| Canton S 50 days | 13 | kk110628 / BSC431[Df] control 40 days 25C | 11 |
| Canton S 70 days | 9 | tinC∆4 > kk110628 / BSC431[Df] 40 days 25C | 11 |
| Canton S 110 days | 14 | Canton S 30 days - vary light | 4 |
| sandman∆ 10 days | 13 | Canton S 40 days - mounting duration | 4 |
| sandman∆ 20 days | 9 | sandman∆ 110 days + EGTA | 3 |
| sandman∆ 30 days | 14 | sandman∆ 110 days + valinomycin | 3 |
| sandman∆ 50 days | 15 | wild-type 30 days - visualize sarcomeres | 9 |
| sandman∆ 70 days | 12 | sandman∆ 30 days - visualize sarcomeres | 4 |

onto a CV-7B headstage (Molecular Devices, Sunnyvale, CA) incorporating an Ag/AgCl electrode and attached to a MP-285 micromanipulator (Sutter Instruments). The data were lowpass filtered to 10 kHz using a Multiclamp 700 B amplifier and digitized at 50 kHz using a Digidata 1440 A analogue to digital convertor and pClamp 10 software (Molecular Devices). Analysis was performed off-line using Clampfit 10.5 (Molecular Devices) and visualized using DataGraph 4$\beta$ (Visual Data Tools). The 200 ms voltage steps in whole cell mode were made in 20 mV increments ranging from −120 mV to +80 mV, from a holding potential of −80 mV. The internal pipet solution contained 150 $K^+$, 5 $Na^+$, 3 $Mg^{2+}$, 161 $Cl^-$, 10 HEPES, pH 7.4 and the bath solution contained 5 $K^+$, 150 $Na^+$, 3 $Mg^{2+}$, 1 $Ca^{2+}$, 163 $Cl^-$, 10 HEPES, pH 7.4 (in mM). Potassium selectivity experiments were also performed in the whole cell mode with an internal pipet solution containing 150 $K^+$, 5 $Na^+$, 3 $Mg^{2+}$, 161 $Cl^-$, 10 HEPES, pH 7.4 (in mM). The bath solution $[K^+]$ and $[Na^+]$ or $[NMDG^+]$ are as indicated in *Figure 5B*, with 3 $Mg^{2+}$, 1 $Ca^{2+}$, 161 $Cl^-$, 10 HEPES, pH 7.4, excepting the 'Divalent-free' solution which substitutes 2 EDTA for the divalent cations (in mM). The superfusion pipette had an internal diameter of 200 µm and the perfusate was gated using an Octaflow multi-valve perfusion system (ALA Scientific, Farmingdale, NY). Each solution was maintained for 10 sweeps of a voltage ramp protocol (200 ms, −120 mV to 80 mV ramp with a 300 ms 0 mV hold). This data was downsampled to 1 kHz and sweeps 2–10 were averaged for each condition, normalized to the peak current observed in the 5 $K^+$ bath solution.

## In vivo pharmacology

110-day-old male flies were imaged before and after bolus injection of 10 mM EGTA or 100 µM valinomycin (Sigma-Aldrich, St. Louis, MO) containing artificial hemolymph-like solution (AHLS): 113 $Na^+$, 5 $K^+$, 8.2 $Mg^{2+}$, 2 $Ca^{2+}$, 133 $Cl^-$, 5 HEPES, 4 $HCO_3^-$, 1 $H_2PO_4{,}^-$, 10 Sucrose, 5 Trehalose, pH 7.5 (in mM). Borosilicate glass pipettes (1 mm OD, 0.75 mm ID, A-M Systems, Sequim, WA) were pulled using a P-1000 puller (Sutter Instruments). Pipette tip diameters of 50–75 µm were created by crushing the taper with forceps and visually confirming their diameter using a microforge (Narishige MF-830). Bolus injections into the abdomen of flies under ice anesthesia were approximately 1000 pL in volume and were made using a Femtojet (Eppendorf, Hamburg, Germany), with the pipette positioned using a manual micromanipulator (World Precision Instruments, Sarasota, FL).

## Intravital imaging of sarcomere dynamics

Cardiomyocyte sarcomere dynamics in the posterior half of the second heart chamber of 30-day-old GFP α-actinin [CC01961] animals (*Buszczak et al., 2007*) were imaged using 19 mW per mm$^2$ blue excitation light, delivered through a 40x/1.3 NA objective (Olympus UPLFLN40XO), with the excitation pattern again restricted to the field of view of the acquired image. Images were captured at 60 frames per second but with otherwise identical camera settings as above. The heart wall detection algorithm was adapted to trace single sarcomeres in YT kymographs of paired left and right cardiomyocytes. The data were subsequently analyzed by calculating the amplitude of coherence between the two sarcomeres:

$$Coherence = -\frac{\partial Position_{Left}}{\partial Time} \cdot \frac{\partial Position_{Right}}{\partial Time}$$

## Acknowledgements

We thank M Petkovic, S Younger, S Barbel and T Cheng for technical support, W Zhang, M Petkovic and S Headland for critical reading of the manuscript, and members of the Jan laboratory for discussion. This work was supported by National Institutes of Health grant (R37NS040929) to YNJ and National Institutes of Mental Health grant (R37MH0653354) to LYJ. MK was supported by a Jane Coffin Childs fellowship and CJP by a junior personnel fellowship from the Heart and Stroke Foundation of Canada. We would also like to acknowledge the *Drosophila* Genomics Resource Center, supported by NIH grant 2P40OD010949-10A1. YNJ and LYJ are investigators of the Howard Hughes Medical Institute.

## Additional information

### Funding

| Funder | Grant reference number | Author |
| --- | --- | --- |
| Jane Coffin Childs Memorial Fund for Medical Research | Postdoctoral Fellowship | Matthew P Klassen |
| Heart and Stroke Foundation of Canada | Junior Personnel Fellowship | Christian J Peters |
| National Institute of Mental Health | R37MH0653354 | Lily Yeh Jan |
| Howard Hughes Medical Institute | Investigators | Lily Yeh Jan Yuh Nung Jan |
| National Institutes of Health | R37NS040929 | Yuh Nung Jan |

The funders had no role in study design, data collection and interpretation, or the decision to submit the work for publication.

### Author contributions

MPK, Conceptualization, Software, Formal analysis, Funding acquisition, Investigation, Visualization, Methodology, Writing—original draft; CJP, Conceptualization, Funding acquisition, Investigation, Methodology, Writing—review and editing; SZ, HHW, Investigation, Methodology, Writing—review and editing; LYJ, YNJ, Conceptualization, Supervision, Funding acquisition, Writing—review and editing

### Author ORCIDs

Matthew P Klassen, http://orcid.org/0000-0002-3633-2512

## Additional files

### Major datasets

The following previously published dataset was used:

| Author(s) | Year | Dataset title | Dataset URL | Database, license, and accessibility information |
|---|---|---|---|---|
| Chintapalli VR, Wang J, Dow JA | 2007 | Using FlyAtlas to identify better Drosophila models of human disease | https://www.ncbi.nlm.nih.gov/geo/query/acc.cgi?acc=GSE7763 | Publicly available at NCBI (accession no. GSE7763) |

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
