## [Decision Letter]

Thank you for submitting your article "Age-dependent diastolic heart failure in an in vivo *Drosophila* model" for consideration by *eLife*. Your article has been reviewed by three peer reviewers, and the evaluation has been overseen by a Reviewing Editor and Naama Barkai as the Senior Editor. The following individual involved in review of your submission has agreed to reveal his identity: Chao Zhou (Reviewer #3).

The reviewers have discussed the reviews with one another and the Reviewing Editor has drafted this decision to help you prepare a revised submission.

Summary:

Klassen et al., describe the development of a novel method to analyze dynamic heart contraction in live adult fly based on high resolution fluorescent imaging. Using this method, they show that, like in humans, the aging *Drosophila* heart manifests primarily defects in relaxation (diastole) rather than in heart contraction (systole). Using their novel method the authors discovered that a pair of two-pore potassium channel (K2P) subunits, largely dispensable in younger flies, are necessary for terminating contraction (systole) in aged animals, where their loss culminates in fibrillatory cardiac arrest. This study paves the road for using the fly heart as a model for understanding the signaling networks maintaining cardiac performance during normal aging.

Essential revisions:

1) Authors need to show that *sandman* and *galene* are expressed in the adult fly heart.

2) Does overexpression of *sandman* "rescue" the aging phenotype?

3) Need a negative control for *galene* – *sandman* interaction in cell culture. Since other K2P channels were screened and had no effects on heart function they should be tested.

Figure 1 – Because it has been shown in previous studies of the fly heart that there are differences in the rates and rhythmicity of anterograde and retrograde hearts beats in intact flies one would expect that heartbeat parameters would also vary. Was the data shown in Figure 1 based on all the heartbeats in a record or subsets of the data binned into anterograde or retrograde? This is not clear nor are the differences in the two types of contractions discussed with respect to the data.

Subsection “Imaging cardiac performance in intact, unanesthetized *Drosophila*”, last paragraph – The authors state that irradiation has no effect on heart function at low levels, however, they cannot actually show what the heart is doing without UV irradiation. In our experience UV irradiation increases heart rate and arrhythmicity. How would they address this?

Subsection “Imaging cardiac performance in intact, unanesthetized *Drosophila*”, last paragraph – Since it has been reported that 80% of female flies are dead after 18 hours of desiccation, it is not clear how the preparation can still be viable after 19 hours glued to a slide.

Subsection “Normal aging manifests as diastolic dysfunction while preserving contractile performance”, second paragraph – The authors group data from the two 'wild-type' genetic backgrounds, *w1118* and *Canton S*, for analysis. This is troublesome as these two genotypes have very different genetic backgrounds – known to influence heart function. The average fly size and heart size in these two genotypes are different. Having similar rates does not seem sufficient rational for grouping data from these two lines. Especially since they are also using this data to generate other parameters that are *not* based solely on rate (e.g. Cardiac output).

Subsection “A pair of K2P subunits, *sandman* and *galene*, are required for terminating systole in aged animals” – Why were two different cardiac drivers used for the *sandman* and *galene* KD? What happens when the two drivers are reversed, does tinCd4>Sandman RNAi have the same effect as for R94C02>Sandman KD? And vice versa for Galene RNAi?

Subsection “Sandman is required to maintain diverse biological oscillators”, first paragraph – "oscillatory balance degenerates catastrophically with age" What does this mean in terms of cardiomyocyte function how does it relate to the persistent contractions in *sandman* KD?

Introduction, third paragraph – The authors misrepresent their system as a high throughput system. If it is high throughput why did they have to pool wt genotypes and analyze only 24 flies per age group for the aging study? In fact, as for all other fly heart model systems including OCT, their system requires manual mounting and orientation of flies and manual tracing of data after the fact to facilitate analysis. These are not characteristics of a high throughput system.

4) The work has one major limitation, however. It is inferred that calcium cycling abnormalities underlie the actual relaxation abnormalities detected, but cardiomyocyte calcium cycling is not measured. This seems curious as the proposed role of calcium is central to the overall proposed mechanism introduced in the last paragraph of the subsection “The *sandman* phenotype reflects a loss of repolarization rather than structurally congestive remodeling”. Moreover, others have successfully measured cardiomyocyte calcium cycling in intact *Drosophila* using fluorescent techniques that should be readily adaptable to the current system (for example, the Maack group's work on *Drosophila* mitochondrial-SR calcium microdomains in Circ Res 2012). Direct calcium measurements are not replaceable by inferences drawn after introduction of EGTA, etc., and their absence here substantially limits confidence in the overall mechanistic conclusions.

---

## [Author Response]

*Essential revisions:*

*1) Authors need to show that sandman and galene are expressed in the adult fly heart.*

We have microdissected the adult fly heart from the rest of the animal and utilized RT-PCR to confirm that *sandman* and *galene* transcripts are indeed expressed there as expected (Figure 3—figure supplement 1).

*2) Does overexpression of sandman "rescue" the aging phenotype?*

We have overexpressed the *sandman* cDNA in the heart of *wild type* and *sandman* mutant animals through 50 days of age, when *wild type* animals show signs of cardiac decline and *sandman* mutants show profound defects in heart function. Overexpression robustly rescues the *sandman* mutant phenotype (Figure 3, Figure 4, and Figure 4—figure supplement 2) but does not appear to improve or degrade cardiac performance in wild-type animals (Figure 3, and Figure 4—figure supplement 2).

*3) Need a negative control for galene – sandman interaction in cell culture. Since other K2P channels were screened and had no effects on heart function they should be tested.*

As a negative control for heteromeric interaction in cell culture, we recorded from CHO cells combinatorially expressing the K2P genes most closely related to *sandman* and *galene*, CG1688 and CG10864, to determine whether these subunits might form functional homomers or, if not, whether they were capable of forming functional heteromeric channels if co-expressed with *sandman* or *galene*. We found that, of all possible combinations of these genes, only *sandman-galene* co-transfection produced measurable currents (Figure 5 and Figure 5—figure supplement 1).

*Figure 1 – Because it has been shown in previous studies of the fly heart that there are differences in the rates and rhythmicity of anterograde and retrograde hearts beats in intact flies one would expect that heartbeat parameters would also vary. Was the data shown in Figure 1 based on all the heartbeats in a record or subsets of the data binned into anterograde or retrograde? This is not clear nor are the differences in the two types of contractions discussed with respect to the data.*

All data combine anterograde and retrograde heartbeats. We do observe qualitative differences in the contraction-relaxation waveform of anterograde versus retrograde heartbeats (e.g. Figure 1), which may reflect differences in pressure waves arising from bolus flow and/or differences in the electrical waveform emanating from the two distinct pacemakers. Although we do plan to extend our computer code to automatically discriminate anterograde and retrograde heartbeats in the future, we do not believe that subdividing our dataset is necessary to support the principal conclusions of our present manuscript. Sustained contractions appear for both directions in *sandman* mutants that are not yet terminally contracted (e.g. Video 2). Furthermore, contraction-relaxation waves become discontinuous and chaotic in *sandman* mutants or *galene* knockdown, which makes any segmentation of anterograde and retrograde heartbeats less trivial than two-point discrimination. Because of these observations, we feel that attempting to subdivide the data into two separate bins would not increase our mechanistic insight and risks diminishing the readability of the manuscript and particularly the figures.

*Subsection “Imaging cardiac performance in intact, unanesthetized Drosophila”, last paragraph – The authors state that irradiation has no effect on heart function at low levels, however, they cannot actually show what the heart is doing without UV irradiation. In our experience UV irradiation increases heart rate and arrhythmicity. How would they address this?*

At the light intensities utilized throughout the paper, we did not observe any meaningful arrhythmicity in wild-type animals. It is possible that spatially restricting illumination to a small region of interest and not using UV excitation has avoided the potential arrhythmic effects previously observed by the reviewer.

We do not use UV irradiation during imaging. We only utilize a brief 10s pulse of 405nm LED light to polymerize the optical glue during cold anesthesia. After 15 minutes of recovery, imaging is performed with green light excitation (538-568nm, 4.3 mW/mm^2^) limited to a small (150x500um) region of interest, which is less than 20% of the total surface of the abdomen. Fly hearts imaged using optical coherence tomography (e.g. Wolf et al., 2006; Alex et al., 2015) or near-IR spectroscopy (Wasserthal, 2007), neither of which utilize UV-activated optical cement, exhibit similar mean heart rates to those reported here (4.5-6.5 Hz vs 5.6 Hz in 10 day animals declining to 3.8 Hz at 110 days).

We tested the effects of increasing green light excitation on the heart and found a 0.39% increase in heart rate per mW/mm^2^ (R^2^ = 0.93) which would suggest that our green light illumination levels increase the heart rate less than 2% (Figure 1—figure supplement 1). To ensure that we minimized any effects arising from the excitation light, we also tested illumination intensities 60% lower than utilized, equivalent to a small spot of restricted light only 50% stronger than sunlight, or less total irradiation than the fly would experience exposed during midday. We observed no differences between these two intensities, suggesting that we minimized any excitation light related artifacts. Light intensities 20-fold higher than utilized exhibited significant tachycardia but only the occasional early aftercontraction.

Subsection “Imaging cardiac performance in intact, unanesthetized Drosophila”, last paragraph – Since it has been reported that 80% of female flies are dead after 18 hours of desiccation, it is not clear how the preparation can still be viable after 19 hours glued to a slide.

We hydrated the immobilized flies overnight using a small drinking capillary and apologize for omitting this detail, now corrected.

*Subsection “Normal aging manifests as diastolic dysfunction while preserving contractile performance”, second paragraph – The authors group data from the two 'wild-type' genetic backgrounds, w1118 and Canton S, for analysis. This is troublesome as these two genotypes have very different genetic backgrounds – known to influence heart function. The average fly size and heart size in these two genotypes are different. Having similar rates does not seem sufficient rational for grouping data from these two lines. Especially since they are also using this data to generate other parameters that are not based solely on rate (e.g. Cardiac output).*

Our conclusion that systolic function is preserved while the kinetics of relaxation are selectively affected during aging is true for both *Canton S* and *w1118* genotypes (see Figure 7). The data were combined because, in our experiments, we found them comparable.

Author response image 1.**DOI:**
http://dx.doi.org/10.7554/eLife.20851.025

The *Canton S* background is the preferred genetic background for behavioral studies, while *white* and particularly *yellow* mutants are known to display behavioral defects in certain assays. We retained the *w1118* allele to facilitate building and confirming transgenic backgrounds but outcrossed it for six generations to our *Canton S* stock to eliminate any unlinked polymorphisms. We collected the *Canton S* and *w1118* datasets to confirm that the retention of *w1118* did not grossly affect cardiac rhythmicity or output. This combined dataset was only used for Figure 2; all other figures utilized the *w1118* dataset alone as all subsequent genetics were performed in that background.

To clarify, all heart parameters are calculated on a heartbeat by heartbeat basis for each fly and do not rely on estimates or parameters generated at the population level in any way. Therefore, if there were material differences in heart function in *Canton S* vs. *w1118* animals, it would manifest in the Tukey boxplots.

If one scrutinizes these datasets separately, they do reveal that the *w1118* dataset has a modest trend towards higher cardiac output (+8.5%) resulting from higher stroke volume, partially mitigated by lower heart rate (see Figure above). The genesis of this difference is unclear but may reflect the fact that our *Canton S* animals are somewhat healthier and are therefore easier to subtly overcrowd during rearing, leading to slightly smaller animals.

It is possible that polymorphisms not linked to the *white* locus have contributed to the differences observed by others, but their genetic identity or prevalence within stocks across the fly community has not been characterized in the literature.

*Subsection “A pair of K2P subunits, sandman and galene, are required for terminating systole in aged animals” – Why were two different cardiac drivers used for the sandman and galene KD? What happens when the two drivers are reversed, does tinCd4>Sandman RNAi have the same effect as for R94C02>Sandman KD? And vice versa for Galene RNAi?*

We utilized tinCd4::Gal4 for all RNAi experiments. It is the strongest heart Gal4 driver and is almost universally used by our peers for heart-specific RNAi (e.g. Neely et al., 2010). We initially used R94C02 for rescue out of genetic convenience given the stocks we had assembled at the time. We have now attempted rescuing the *sandman* phenotype using two additional Gal4s to allay the concerns expressed by the reviewer. tinCd4::Gal4 robustly rescues the *sandman* phenotype while a Gal4 expressed in many muscle subtypes excepting the heart (R37B05::Gal4) does not rescue (Figure 4—figure supplement 3).

*Subsection “Sandman is required to maintain diverse biological oscillators”, first paragraph – "oscillatory balance degenerates catastrophically with age" What does this mean in terms of cardiomyocyte function how does it relate to the persistent contractions in sandman KD?*

The term “oscillatory balance” was meant to reflect the balance between contraction and relaxation that progressively trends toward persistent contractions in aged *sandman* mutants. This is most easily observed in Figure 4—figure supplement 1, where the median heart contraction is not balanced with an equivalent magnitude heart relaxation in aged *sandman* mutants. We have clarified this language in the manuscript.

*Introduction, third paragraph – The authors misrepresent their system as a high throughput system. If it is high throughput why did they have to pool wt genotypes and analyze only 24 flies per age group for the aging study? In fact, as for all other fly heart model systems including OCT, their system requires manual mounting and orientation of flies and manual tracing of data after the fact to facilitate analysis. These are not characteristics of a high throughput system.*

Our apologies. In our defense, we routinely mount and record 15 flies, collecting approximately 7,000 heartbeats per hour, and our heart wall detection and heartbeat segmentation algorithms are fully automated. We believe this is a significant throughput advance over existing methodologies for any heart system. The only manual steps are selecting the longitudinal position for the kymograph and tracing the chamber once to estimate its average systolic volume, which is script assisted and takes less than 30 seconds per fly.

The wild-type dataset comprises 146 flies in total. We combined the *w1118* and *Canton S* datasets because they were not meaningfully different from one another and were available for inclusion. We did not collect additional flies for this dataset as the principal conclusions were readily quantifiable at this sample size. We strongly believe that our methodology scales particularly well and will permit considerable forward genetic screening in future studies but have moderated references to throughput in line with the reviewer’s comments.

*4) The work has one major limitation, however. It is inferred that calcium cycling abnormalities underlie the actual relaxation abnormalities detected, but cardiomyocyte calcium cycling is not measured. This seems curious as the proposed role of calcium is central to the overall proposed mechanism introduced in the last paragraph of the subsection “The sandman phenotype reflects a loss of repolarization rather than structurally congestive remodeling”. Moreover, others have successfully measured cardiomyocyte calcium cycling in intact Drosophila using fluorescent techniques that should be readily adaptable to the current system (for example, the Maack group's work on Drosophila mitochondrial-SR calcium microdomains in Circ Res 2012). Direct calcium measurements are not replaceable by inferences drawn after introduction of EGTA, etc., and their absence here substantially limits confidence in the overall mechanistic conclusions.*

Unfortunately, there are no published methodologies for measuring cardiomyocyte calcium cycling in intact *Drosophila*. We agree that it would be useful to monitor the calcium and voltage waveforms in vivo but the field has yet to surmount technical limitations associated with these experiments. The calcium wave observed by the Maack group and others utilizes the in situ preparation which beats at approximately 1-2Hz and is therefore sufficiently slow for GCaMP calcium imaging. However, this calcium wave is several fold longer in duration than what one would expect to observe in vivo, which may arise from the heart being removed from its normal physiochemical environment and/or slow GCAMP off kinetics distorting the signal. The fly heart beats at approximately 5-6 Hz in vivo, which means that best current generation genetically encoded calcium indicators do not have sufficiently rapid off-decay times (t_1/2_ GCaMP6f 350ms, Chen et al., Nature 2013) to adequately resolve individual calcium waves in vivo. Given such dramatic differences in the calcium waveform in situ vs. in vivo, it stands to reason that there exist significant differences in the relative strength and kinetics of the various inward and outward conductances, making comparisons about ion channels problematic.

We invested considerable time attempting to overcome this technical hurdle, even testing GCAMP6 variants with reduced calcium affinity, but have determined that the temporal resolution of the optical signals is currently inadequate, compounded by the difficulty imaging cytosolic GFP signals intravitally and addressing motion artifacts that cannot be deconvolved without ratiometric approaches. This last point of particular concern as most published papers using this methodology neglect to compensate for motion artifacts arising from the contraction-relaxation cycle, which Lin et al., 2011, who developed this methodology for the *Drosophila* heart, demonstrated represents 50% of the resultant signal (see their Supplementary Figure 1). Effectively, one can derive an equivalent ‘calcium’ signal using cytosolic GFP. Cetoxymethyl ester conjugated calcium sensitive dyes, while having sufficiently fast off-kinetics and internal ratiometric controls, have been reported by others to load very poorly in *Drosophila* (e.g. Lin et al., 2011 and our experience). We tried paralyzing the heart *in situ* using blebbistatin and/or cytochalasin-D but have found that these compounds trigger significant dysrhythmia in wild-type animals, as measured by genetically-encoded voltage indicators.

While we are sympathetic to the reviewer’s desire to directly visualize calcium, we feel that the experiments detailed in Figure 6 should not be discounted as inadequately supporting our model as it relates to the scope of this publication. We are unaware of any mechanistic basis for a calcium independent contraction-relaxation of the sarcomere but have moderated our text accordingly.

To better understand the terminal phenotype of *sandman* mutants, we directly visualized sarcomere dynamics in vivo, for which the biophysical linkage between calcium and contraction is well established. This was a significant technical accomplishment and provides excellent spatiotemporal resolution of individual sarcomeric behavior (Video 3). During the sustained contractions observed in *sandman* mutants, sarcomeres actively contract and relax out of phase with one another, suggesting that sufficient ATP was available locally to drive myosin activity, myosin dissociation and the various pumps that can sequester calcium. Interestingly, this fibrillatory behavior periodically and rapidly resolves itself in a synchronous manner globally (Figure 6, Video 3), an effect that is most plausibly explained by a global repolarization closing sarcolemmal Ca_v_ channels, which inactivate much more slowly in *Drosophila* relative to vertebrates (t_1/2_ 1-2s, Haraet al., J Neurogenetics 2015) and could therefore generate a prolonged window current sustaining calcium influx in the absence of sufficient repolarization back to the resting potential.

Further support of our model comes from our pharmacological experiments, which importantly are acute (seconds). First, chelating extracellular calcium using membrane impermeable EGTA demonstrates that calcium influx across the sarcolemmal membrane is necessary for sustaining contractions in *sandman* mutant animals. Second, creating an artificial but highly selective K^+^ leak across the sarcolemmal membrane using valinomycin also terminated the persistent contractions. Accordingly, these experiments exclude a calcium and/or potassium independent means of sustaining the persistent contractions in sandman mutants. Because potassium channels play well-established roles in cardiac repolarization back to the resting potential and L-type voltage gated Ca channels are essential for heart contractions in *Drosophila*, we feel that a mechanistic link between the loss of a potassium channel and the persistent contractions observed is not unduly speculative.

What we do not yet understand is why the phenotype is progressively age-dependent and what mechanisms cause the fibrillatory cycling of individual sarcomeres during the sustained contractions. Having high-resolution voltage indicators as well as calcium measures for the cytosol and SR would allow us to address these questions, which we hope to develop in a future manuscript. In conclusion, we feel that our intravital imaging methodology and our characterization of this novel potassium channel are worthy of publication and look forward to extending this work in future studies.